# Parent Mobile Phone Use in Playgrounds: A Paradox of Convenience

**DOI:** 10.3390/children7120284

**Published:** 2020-12-10

**Authors:** Keira Bury, Jonine Jancey, Justine E. Leavy

**Affiliations:** 1School of Public Health, Curtin University, Perth 6845, WA, Australia; keira.bury@curtin.edu.au; 2Collaboration for Evidence, Research and Impact in Public Health, School of Public Health, Faculty of Health Sciences, Curtin University, Perth 6845, WA, Australia; j.jancey@curtin.edu.au

**Keywords:** social determinants, children, child-play, mobile phone, injury, supervision

## Abstract

Creating social and physical environments that promote good health is a key component of a social determinants approach. For the parents of young children, a smartphone offers opportunities for social networking, photography and multi-tasking. Understanding the relationship between supervision, mobile phone use and injury in the playground setting is essential. This research explored parent mobile device use (MDU), parent–child interaction in the playground, parent attitudes and perceptions towards MDU and strategies used to limit MDU in the playground. A mixed-methods approach collected naturalistic observations of parents of children aged 0–5 (*n* = 85) and intercept interviews (*n* = 20) at four metropolitan playgrounds in Perth, Western Australia. Most frequently observed MDU was scrolling (75.5%) and telephone calls (13.9%). Increased duration of MDU resulted in a reduction in supervision, parent–child play and increased child injury potential. The camera function offered the most benefits. Strategies to prevent MDU included turning to silent mode, wearing a watch and environmental cues. MDU was found to contribute to reduced supervision of children, which is a risk factor for injury. This is an emerging area of injury prevention indicating a need for broader strategies addressing the complex interplay between the social determinants and the developmental younger years.

## 1. Introduction

For the parents of young children, a mobile device, such as a smartphone, offers opportunities for communication, social networking, photography, personal organisation, work flexibility and multi-tasking [1,2,3,4]. However, it can also be a source of distraction that brings about feelings of guilt and concern for poor role modelling of mobile device use (MDU) for young children during their early development [1,4,5,6,7,8]. Parent MDU can have detrimental effects on family relationships including interaction and attachment [3,5,9,10], reducing parents’ responsiveness when children try to attract their attention [3,5] and social support [11]. 

The early years of life represent a vital time during which lifelong trajectories for health are determined by the complex interplay between the social determinants [12]. Social determinants of health are conditions in the environments in which children are born, live, learn, interact and play [13]. Creating social and physical environments that promote good health is a key component of a social determinants approach to health [14]. The social environment includes interactions with family and parent–child play encompasses social relationships in physical settings such as playgrounds [12,14,15]. Of interest, outdoor play is deemed fundamental for the physical, cognitive, emotional and social well-being of children [15,16], and has been declared a right of every child by the United Nations Convention on the Rights of the Child [17]. Outdoor play in early childhood has been linked to the social determinants that shape health and development, emphasising children’s participation in play is influenced by multiple interacting social and contextual factors [15]. Very few studies have focused on understanding the social (e.g., parental supervision) and contextual (e.g., playground setting) factors that may influence behaviours and the relationship with the injury. 

Whilst parent supervision is recognised as being a protective factor for injury in young children [18,19,20], there is evidence that there is a reduction in supervision quality as a result of parent MDU [21,22], and an association between parent MDU and injury in young children [23]. A recent US study found that children’s safety needs were more often at risk when parents used their phones than when they were distracted by other factors, such as talking with other adults or reading printed material [24]. A review of the impact of parent MDU on parent–child interaction found increased phone ownership and related increases in parents’ lack of attention to children could be associated with increases in childhood injuries [25]. US research has also reported a positive association between young children presenting at hospitals with injuries and the rapid adoption of smartphone use [23,26]. One study concluded that distraction from increased mobile phone use was directly associated with an increase in emergency hospitalisation for injuries in children aged 0–5 years; however, there was no association with children aged 6–10 years [23]. Hiniker et al. (2015), on the other hand, reported parents did not use their phone in the playground setting because they believed it would compromise their child’s safety and lower their ability to respond. There is an increasing interest in understanding the impact of mobile device use by parents/carers of children aged 0–5 years [1,25], which commands an investigation into which characteristics of MDU are most absorbing of their attention [9,25,27] when supervising children. 

The aim of the naturalistic study was twofold: first, to observe mode and duration of parent and caregiver MDU (telephone call, scroll/type or camera), parent/caregiver interaction and the coinciding behaviour (specifically: supervision, interaction and child injury) in the playground setting. Second, to explore parent/caregiver attitudes and perceptions of MDU and strategies used to limit MDU in the playground setting. 

## 2. Materials and Methods

### 2.1. Study Design

A mixed-methods approach was used to collect quantitative and qualitative data from parent or caregiver/child dyads in playgrounds in Perth, Western Australia (WA). Trained researchers worked in teams of two to conduct covert, naturalistic observations (*n* = 85). In addition, intercept interviews were conducted with parents/caregivers (*n* = 20). The sample size for the observations was informed by naturalistic observational methodologies for data collection conducted in New South Wales, Australia (*n* = 50) [4] and New Jersey, US (*n* = 60) [27], whilst one other study in the US collected 171 observations over almost double (seven parks) [24] compared with four in this study. The sample size for the interview numbers (*n* = 20) were consistent with similar studies [4,27].

### 2.2. Study Setting

The study was undertaken in four playgrounds across a range of Socio-Economic Indexes for Areas (SEIFA) [28] in metropolitan Perth, WA. The playgrounds included: (i) a fenced playground with several different play areas, built specifically for 6-year-olds and under; (ii) a fenced playground with low-level equipment for toddlers; (iii) an ungated single playground adjacent to a number of BBQ pits, and grass/benches for picnics; and (iv) an ungated single playground. Observations were undertaken on school days between the hours of 9:30 a.m. and 12:30 p.m., a time when older siblings have been dropped off at school and a popular time for 0–5 years to attend playgrounds [4,27]. 

### 2.3. Study Participants

Participants were parent or caregiver/child dyads where the parent/caregiver appeared to be under 45 years, attending the playground by themselves (not with another adult) and caring for at least one independently mobile child who appeared to be under 5 years. The estimation of the child’s age was based on general appearance and height. Parents caring for more than one child were included in the study. Where parent/carers had multiple children under their care, the researchers chose the child that best fitted the criteria—under 5 years and independently mobile. Participants were not notified they were being observed so as to obtain naturalistic data. Participants who left the playground or were lost from the site (into a café/toilet) within the first 10 min of the observation were excluded from the study. Researchers recruited participants for intercept interviews (*n* = 20). Parent/Carers appearing to meet the participant criteria were approached and invited to participate in the study. No parent/carers declined to be interviewed. Participants were provided with an information sheet and consent form prior to the commencement of interviews. Ethics approval was obtained for this study by Curtin University Research Ethics Committee no: HRE2016-0027.

### 2.4. Quantitative and Qualitative Data Collection

#### 2.4.1. Quantitative

The two researchers positioned themselves within the boundary of the playground to observe participants, one observation at a time. At the beginning of the observation period, the researchers synchronised stopwatches to concurrently record data for a period of up to twenty minutes. Participants were observed for MDU (mode and duration of use), caregiver supervision, caregiver/child interaction and child injury potential with data recorded during each minute. 

The first researcher collected data for caregiver MDU using an adapted, previously trialled mobile device timing audit [4,27]. During each minute of the observation, the researcher observed the dominant mode of caregiver MDU (telephone call, scroll/type or camera) and the duration of caregiver MDU (recorded as a minimum of 0 s and a maximum of 60 s). The summed duration for each minute of observation was later categorised as; “No MDU”; “1–10 s”; “11–20 s”; “21–30 s”; “31–40 s”; “41–50 s” and “51–60 s”. The second researcher collected data on caregiver supervision, caregiver–child interaction and child injury potential using a combined observation audit [4,27]. During each minute of the observation, the researcher observed and recorded the dominant behaviour. 

#### 2.4.2. Qualitative

Caregivers appearing to meet the participant criteria were approached and invited to participate in the study. Participants were provided with an information sheet and consent form prior to the commencement of interviews. The researcher recorded the intercept interviews using a structured interview schedule. Each interview took approximately 10 min to complete. The interviews were conducted by two researchers, field notes were kept by both researchers and compared for consistency at the completion of each interview period.

### 2.5. Measures

Data from the mobile timing device audit were collapsed and used to create three categorical variables:(1)MDU: this binary variable categorised minutes as either “No MDU” (0 s of MDU) or “MDU” (1–60 s of MDU).(2)MDU Mode: this variable categorised minutes as “Telephone call” (using the device telephone call function) “Scroll/type” (using the device touchscreen function), or “Camera” (using the device camera function).(3)MDU Duration: this variable categorised the summed MDU duration for each minute as “No MDU” (0 s of observed), “1–10 s”, “11–20 s”, “21–30 s”, “31–40 s”, “41–50 s”, “51–60 s” (of observed MDU).

Data from the caregiver–child observation tool for supervision, interaction and injury potential were collapsed:(1)Supervision: “Constant supervision” (caregiver watching, following, mediating, redirecting the child or remaining in close proximity); “Intermittent supervision” (caregiver sought visual contact with the child intermittently); “No supervision” (caregiver had no contact with the child).(2)Interaction: “Caregiver-child play” (caregiver and child play together), “Independent play” (child plays without caregiver), “Verbal interaction” (caregiver and/or child talking or calling to each other), “Hold/touch” (physical contact between caregiver and child), “Sitting/eating/drinking” (caregiver and child in close proximity having a drink or food).(3)Injury Potential: Child injury risk was measured using an adapted injury risk behaviour checklist developed by Dotson (2013). Injury potential was categorised as “Increased injury” potential (unsafe play behaviours, i.e., child passes within moving radius of equipment, child uses equipment in an unintended manner), “Decreased injury” potential (safe play behaviours, i.e., child takes precaution, child stops swing and dismounts) or “Inadequate carer supervision” (i.e., child moves out of view of caregiver, the caregiver does not give child direction on how to behave safely).

### 2.6. Analysis

#### 2.6.1. Quantitative Data 

The data for the 85 observed participants were broken down into one-minute blocks which resulted in a total of 1532 min of observation time which was entered into SPSS version 23 [29]. Descriptive statistics were used to describe the parent/caregiver participant characteristics and the children under the parent/caregiver’s supervision. The quantitative analysis explored the coinciding MDU, supervision, interaction and injury risk behaviours within each minute of the observation. A series of cross-tabulations with Pearson chi-square tests with a significance level of *p* < 0.01 were conducted to test the association between MDU, MDU Mode, MDU Duration and the outcome variables Supervision, Interaction and Injury Potential. 

#### 2.6.2. Qualitative Data Collection and Analysis

The interviews were transcribed verbatim by one researcher (KB) and were divided among the research team for open coding (KB, JL, JJ). The first researcher then collated the coded interviews using NVIVO 12 [30] and identified relationships between the codes to form the emerging themes which were agreed upon by the research team. Data saturation was reached during the data collection process (*n* = 20) and no new concepts emerged. Identified themes were consistent with work previously undertaken by Australian researchers [4]. The general inductive approach is a straightforward easily used, systematic set of procedures for analysing qualitative data and provides reliable findings [31]. Participant quotes to support the themes have been de-identified and presented in the results.

## 3. Results

### 3.1. Playground Observations

#### 3.1.1. Participant Characteristics

Participants were mostly female (*n* = 72, 85%) and caring for one child under the age of 5 years (*n* = 68, 80%). Of the 85 parent/caregiver–child dyad observations (*n* = 85), 47 were in high socio-economic status (SES) playground locations and 38 in low SES playground locations. 

#### 3.1.2. Mobile Device Use—Mode and Duration 

During the observation period, caregiver MDU was observed among 70% (*n* = 59) of the caregiver–child dyads, 30% (*n* = 16) of parents/carers did not use their mobile device, with total caregiver MDU comprising 23.5% of the observed time. The most frequently observed mode of MDU was “Scrolling/typing” (*n* = 272, 75.5%), followed by telephone call (*n* = 50, 13.9%) and using the camera (*n* = 38, 10.5%). The duration of MDU was either short of “1–10 s” (*n* = 91) or almost a full minute (“51–60 s”) (*n* = 92). 

#### 3.1.3. Caregiver Mobile Device Use and Caregiver Supervision

Table 1 presents a comparison of supervision behaviours between minutes of “No MDU” and minutes of “MDU”. For all observed minutes, the proportion of “No supervision” was higher when using a mobile device (MDU) compared with “No MDU” (29.4% and 5.2%, respectively). The proportion of “Constant supervision” was similar for “MDU” (45.8%) and “No MDU” (43.6%). A significant association was found between MDU and caregiver supervision χ^2^(2) = 191.67, *p* ≤ 0.001.

#### 3.1.4. Caregiver Mobile Device Use and Caregiver–Child Interaction 

“Independent play” (32.6%, *n* = 500) and “Caregiver–child play” (35.4%, *n* = 542) each made up one-third of the interaction behaviour of all observed minutes (Table 2). However, the proportion of “Caregiver–child play” with MDU was half that of no MDU (20.3% and 40.0%, respectively), while the proportion of “Independent play” for MDU was double that of “No MDU” (50.3% compared with 27.2%, respectively). The results show similar proportions for “Verbal interaction” within the “MDU” and “No MDU” groups (22.2% and 20.8%, respectively). A significant association was found between MDU and caregiver–child interaction χ^2^(4) = 81.95, *p* ≤ 0.001.

#### 3.1.5. Caregiver Mobile Device Use and Child Injury Potential 

Increased child injury potential was observed among 9.5% (*n* = 146) of all observed minutes. The proportion of observed “Increased injury potential” for “No MDU” was 5.7% (*n* = 67) compared with 21.9% (*n* = 79) with “MDU” (Table 3). The highest proportion of “Increased injury potential” was observed in the “51–60 s” (*n* = 30, 32.6%) MDU duration category. When comparing child injury potential across the mode of caregiver MDU, the highest proportion of “Increased injury potential” was observed for “Telephone call” (*n* = 13, 26%). 

### 3.2. Playground Interviews 

#### 3.2.1. Participant Characteristics 

Interviewed participants (Table 4) were mostly female (*n* = 15, 75%), born in Australia (*n* = 12, 60%), university educated (*n* = 14, 70%) and supervising one child (*n* = 14, 70%) with a mean age of 2.5 years. 

#### 3.2.2. Reported MDU Behaviours

Almost all participants reported using their mobile device at the playground that day (*n* = 19, 95%). Participants made short statements around their reason for their MDU in the playground on that day such as “*Had to answer a couple of business calls*” (P13) and “*Just to check if I’ve received any messages*” (P21). The most frequently reported reason for MDU included methods of communications (calls, text, email) (*n* = 12, 60%). Other reasons included photography (*n* = 9, 45%) and using the device to undertake specific tasks for personal administration (*n* = 3, 15%). 

#### 3.2.3. Perspectives on Caregiver MDU 

Four dominant themes emerged from the participant interviews; these were: convenience, distraction, security and making memories. 

#### 3.2.4. Convenience 

Participants cited essential daily communication including telephone calls and text-messaging, connecting through social media networks, easy access to information, and photography, for example, “*If you need to know where a street is you can just google it. Taking photos of my child, sending them to different people would be an advantage and most importantly, if there’s an emergency I can call someone*.” (P19).

In addition, other activities were web browsing for news, sports scores, attending to work-related matters, or activities for personal “admin”, such as banking, shopping and coordinating services such as car repairs. The ability to undertake these activities anytime, anywhere *“I guess you can multitask, you can get those little things done when you have a chance”* (P20) contributed to the convenience and the need to carry a mobile device. 

#### 3.2.5. Distraction

In contrast, most participants reported that notifications, social media and the obligation to check in with work were a distraction, for example, “*If you’ve got your phone it’s very easy to be distracted and to keep it constantly in use and to be distracted from doing this I’ll just check Facebook.”* (P2) Whilst the obligation to check in with work was also a distraction, *“You can feel obligated to check work emails, like I’m just stepping out of work for today just to watch him for a minute but you do feel obligated to be checking, too available.”* (P5).

Participants highlighted a paradox between the convenience and distraction of MDU, giving reasons why they would not use their device in the playground. For example:

“*I guess that you have to be conscious that it doesn’t distract you from spending this time with the kids…, and giving them your full attention because you want to be interacting, you want to be there with them in the moment*”. (P20) However, there was a tension between distraction and convenience, “*It’s like a psychological kind of thing where I’m drawn to answer that beep and that takes me away from my kid. It’s on silent now. It’s just a distraction. I use it for my convenience rather than other people’s convenience*.” (P16).

Participants believed that brief moments of MDU, such as making a telephone call would have the least impact on supervision, whereas lengthy periods viewing the device for text messaging or social media were more likely to diminish the quality of supervision, “*I think if you’re scrolling through Facebook, then that obviously requires a lot more attention than if you’re communicating on a phone. I think you’re much more able to supervise if you’re only using it in its basic sense whereas if you’re typing a text or you’re writing an email, scrolling through Facebook then I think your ability to supervise is impaired. Coz obviously it takes more concentration*.” (P5).

Using the camera was viewed differently “*Other than photos, anytime you are with your kids you should be supervising them so in public unless you’ve got somebody else with you who can take over that role. I always have eyes on (child) even through the camera on my phone. I think that’s the main thing*.” (P16).

Participants commented on the social acceptability of MDU within the playground setting with work tasks or essential telephone calls deemed more acceptable distractions than social media, along with some criticism of this behaviour. For example, “*There’s not too many things that are more important than watching a kid, you know if he’s close to the road you don’t need to use your phone but if he’s in the playground and I need to take a phone call for any particular reason. But I don’t think there is any reason that you need to use Facebook or something like that if the kid needs watching*.” (P5).

#### 3.2.6. Security

Participants frequently reported the benefit (*n* = 14, 70%) of having their mobile device in case of an emergency within the playground, “*The phone is just there as a secondary device as an emergency or if you need to contact someone.*” (P21); or outside the playground setting, *“I really just try not to answer your phone unless you absolutely have to. You don’t want to miss a call either, if it’s something important like family or an emergency.*” (P18). 

When considering supervising children, participants made comparisons to other settings, for example, being in and around water, “*She actually just turned around to talk to me for a few seconds and that’s all it took for a one year old to nose dive into the pool…I think so much can happen in the blink of an eyelid, you can’t take your eye off them*.” (P18). Or other settings like driving a car, “*You can’t have your eyes on two things at once, it’s impossible. Other people maybe can, I don’t think I can. It’s like the driving, they wouldn’t have it banned if it were safe. Surely*.” (P22).

#### 3.2.7. Making Memories 

Photography was highlighted as a valuable aspect of MDU which enhanced their experience at the playground, *“To capture precious moments, to be in contact with other people that aren’t with you. Family overseas, you can post a photo of Facebook they can kind of be a part of it as well.”* (P10) It enabled them to capture memories with their children and share these with loved ones at a later date. 

Caregivers were concerned about the ability to interact with and supervise children while using a mobile device, particularly if it involved using social media. However, photography was commonly indicated as an acceptable mode of MDU in the playground indicating the perception that photography can both mitigate safety concerns and enable interaction with children. For example: “*I’m supposed to be looking after her, engaging with her, supervising her and making sure that she’s safe unless it’s for the camera*.” (P16).

#### 3.2.8. Strategies for Limiting MDU around Children

Caregivers were forthcoming about their desire to limit MDU, which stemmed from wanting to spend quality time with their child, avoid distraction in risky settings, and role model appropriate technology use, so as to not raise device focussed children. 

All participants nominated strategies to limit their MDU use in the playground setting, with the most common strategy defined as abstinence, “*Don’t bring it out, don’t bring it, wear a watch!*” (P6). Other participants, however, referred to restricting the functionality of their device by putting it on silence or only using it for a camera, “*Put it on silent. I don’t have my phone on when I’m meant to be watching her because I know how quickly things can change as well. Also having it on silent I don’t kind of have that niggling feeling to look at all the beeps that go off. Just think of it as a camera when you’re at the park, nothing else*.” (P16).

Participants reported ‘downtime’, which were times when they perceived their child was occupied and required less supervision or interaction. One participant explained “*Most things require you to look at the screen. When she’s on the swing. I’m not as concerned, coz she’s strapped in, she’s holding on. I look at it frequently, don’t get me wrong, I’m not some saint.*” (P15).

Some participants reported the need to role-model positive MDU behaviours for their children, “I don’t really think they need to come out of your bag when you’re at the playground or at schools or in social situations with your own friends, you should be engaging with the people there in front of you. Family mealtimes, it’s really important to sit down together, have dinner, have a discussion. I think kids today will miss out on a lot of social skills if they see everyone on their smartphone all the time.” (P19). This resulted in curbing their MDU. 

## 4. Discussion

This naturalistic study explored mobile device use by parent/caregivers of children aged 0–5 years in four playgrounds in metropolitan Perth, Western Australia. The study found that parent texting/scrolling, telephone calls and camera use were a common occurrence in the playground setting (70% of observation participants) but occupied a small part of their time at the playground. This finding is consistent with other research in Australia and the US [4,27], however, the paradox between convenience and distraction was a salient theme of the study which has also been found in other research [7,24,32,33]. Of interest, parents and carers suggested that the practicality of having a mobile phone close-by was reassuring in the event of an emergency, thus creating a tension between having the mobile phone close by or not at all. Child-injury prevention agencies and playground designers should explore broad strategies to promote the importance of supervision to prevent child injury, reduction in MDU in social settings and the promotion of parent–child play. Going forward, policies and programmes must embrace the social determinants of health that play a vital role in the development years. 

It has been suggested that the longer the time spent on a mobile device, increases the negative impact on supervision, and decreases interaction with the child [24]. This finding was supported by our observations, for as the time on the mobile device increased, the supervision and caregiver–child play decreased, being replaced by no parent/carer supervision, and independent play by the child. Furthermore, when parents were on the mobile device for a full minute, the potential for injury increased, compared with those parent/carers not using their mobile device. The injury potential increased when the supervision was interrupted by scrolling and increased again when on a telephone call. There is recent evidence suggesting children’s safety needs are compromised when parents use their mobile phones [34], and smartphone ownership by parents’ may help explain the increase in young children presenting at ED with injuries [23,26]. Interviewees acknowledged the potential for interrupted supervision, especially if there was an opportunity to multi-task (e.g., telephone calls or check work emails). However, worthy of exploration is the notion that a parent/carer may take the child to the playground more frequently because they can use their smartphone to multi-task, e.g., access e-mails, and in turn the child may get injured more, not because the caregiver is distracted but simply because the child goes to the playground more [23]. Our findings support a recent review where parents reported the need for uninterrupted supervision of children in and around playgrounds, roads and waterways where there is the potential for childhood injury [25]. Further research to fully understand the level of device distraction, multi-tasking and childhood safety to provide guidelines for parents’ use is a research priority.

This study found that the tasks (telephone call, scroll/type or camera) undertaken on the mobile device influenced supervision and the opportunity for the social interplay between parents and their children. Participants reported that scrolling/texting resulted in a break from visual supervision, however, using the camera function not only maintained supervision but was often associated with play interaction between parents and their child. Of interest, parent/caregivers did not connect camera use with being distracted, and this finding is consistent with another recent Australian study that found parents value mobile devices as a way to capture memorable moments [4]. Noteworthy, the longer the time spent on taking or making a telephone call was found to be most closely associated with no supervision and minimal play between the parent and the child. Parental behaviours have a critical role as a social determinant of child development. For example, positive reinforcement while on play equipment, displays of warmth and affection result in fewer child behaviour problems and positive peer relations that, in turn, enhance a child’s health [12]. An investment in early child development as a determinant of health will translate to learning skills and increased well-being across the life course.

Using a social determinants lens, early childhood development opportunities are affected by various social and environmental factors, including relationships with parents and caregivers [12]. The playground, specifically controlled, gated playground environments purpose-built for those under 6 years of age, provide co-benefits for the child and parent/carer. There is an opportunity for children’s involvement in independent, active and risky play [34,35,36]; and a time for parents to ‘take a break’ from constant supervision acknowledging that some minor injuries are an inevitable aspect of early childhood [37]. As such, factors including a child’s age, physical ability and the playground design have the potential to influence the level and type of supervision, and the benefit of parents using their mobile device whilst at the playground. Furthermore, recent literature supporting risky play, i.e., a play that is thrilling and exciting but includes the possibility of physical injury, has been positively associated with physical and social health in children [36]. However, despite the differing designs of the four playgrounds in this study, ranging from a large gated setting with multiple play areas to a small single ungated play area, similar patterns of MDU and supervision were observed. Further exploration of the mitigation of potential risks of MDU and the interplay of the benefits might be an area worthy of future research.

Social acceptability and parent role-modelling were identified as influences of parent MDU. Quick calls, text messages and work were deemed more acceptable than using the device for social media or entertainment. Observations of other adults MDU in the playground were reported to influence interviewee’s attitudes to their own MDU and how this relates to ‘good parenting’ but also the importance of role-modelling non-intentional use [27] to their children and to other adults in the playground setting. These perspectives contrast to prior work where caregivers were less critical of others and reserved judgement about the appropriateness of others’ MDU [27]. These findings may highlight a changing social climate that is becoming more critical of parent/caregiver MDU and may indicate a need for realistic population-wide strategies to raise awareness of the risks associated with the distraction caused by mobile phones.

Most interviewees indicated that they had experienced success in limiting their own MDU and offered a range of strategies to help minimise use when caring for children. These strategies included abstinence, achieved by leaving the device in a bag, at home or in the car, choosing a phone function mode such as a silent or camera, and synchronising MDU during times when they perceived children required less interaction or supervision. In addition, a less often mentioned but observed strategy was caregiver proximal supervision i.e., following the child around the playground and keeping close whilst using a mobile device. This behaviour enabled the caregiver to maintain ‘Constant supervision’ whilst engaging in MDU [20]; however, it did not support caregiver–child interaction and exploit the time for parents and children to interact in a play setting. 

### Strengths and Limitations

To our knowledge, this is the first research to examine an association between the characteristics of parent/caregiver MDU (duration and mode) and caregiver–child interaction, caregiver supervision and child injury potential, providing new evidence. A mixed-method approach provided a range of perspectives and strategies that parents use to limit their MDU around children. There were a number of limitations including: an over-representation of females; small sample size, however, this is consistent with other similar research [4,24,27]. The approximate age of the child in the parent/child dyad was not estimated (other than being under 5 years), so we did not have an approximated age break down to complete any additional analysis by age, and half of the playgrounds being located in higher SES, metropolitan areas. 

## 5. Conclusions

The majority of caregivers reported that whilst MDU was convenient for communication, personal organisation and security, it was an unwanted distraction in the playground, where supervising and interacting with children should be the priority. However, the exception was the camera, which was highly valued by caregivers for making memories and also offered the most support for maintaining supervision and interaction through play. Mobile device use was found to contribute to reduced supervision of children, which is a risk factor for injury. This is an emerging area of injury prevention that indicates the need for broader strategies addressing the complex interplay that exists between the social determinants and the developmental younger years. Finally, the caregiver perspectives from this research are valuable for the development of realistic and effective strategies that support parents and caregivers to achieve their desired MDU. 

## Figures and Tables

**Table 1 children-07-00284-t001:** Caregiver supervision by minutes of no mobile device use (No MDU) compared with minutes of mobile device use (MDU).

Caregiver Supervision *	No MDU		MDU		Total	
	*n*	%	*n*	%	*n*	%
Constant supervision	511	43.6	165	45.8	676	44.1
Intermittent supervision	600	51.2	89	24.7	689	45.0
No supervision	61	5.2	106	29.4	167	10.9

* *p* ≤ 0.001.

**Table 2 children-07-00284-t002:** Caregiver–child interaction by minutes of no mobile device use (No MDU) compared with minutes of mobile device use (MDU).

Caregiver–Child Interaction *	No MDU		MDU		Total	
	*n*	%	*n*	%	*n*	%
Caregiver–child play	469	40.0	73	20.3	542	35.4
Verbal interaction	244	20.8	80	22.2	324	21.1
Hold/touch	75	6.4	11	3.1	86	5.6
Sitting/eating/drinking	65	5.5	15	4.2	80	5.2
Independent play	319	27.2	181	50.3	500	32.6

* *p* ≤ 0.001.

**Table 3 children-07-00284-t003:** Child injury potential by minutes of no mobile device use (No MDU) compared with minutes of mobile device use (MDU).

Child Injury Potential	No MDU		MDU		Total	
	*n*	%	*n*	%	*n*	%
Decreased	1105	94.3	281	78.1	1386	90.5
Increased	67	5.7	79	21.9	146	9.5
	1172	100.0	360	100.0	1532	100.0

**Table 4 children-07-00284-t004:** Demographic characteristics of interview participants (*n* = 20).

Characteristic	*n*	%
**Sex**		
Female	15	75
Male	5	25
**Age (years)**		
25–29	4	20
30–34	9	45
35–39	3	15
40+	4	20
**Country of birth**		
Australia	12	60
Other	8	40
**Highest level of education**		
Year 12 or Equivalent	1	5
Trade/Diploma	5	25
Bachelor Degree or higher	14	70
**Working status**		
In paid work	10	50
Not in paid work	10	50

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
