# Peer review of "Parent Mobile Phone Use in Playgrounds: A Paradox of Convenience"

_children, 2020, doi:10.3390/children7120284_

Round 1

Reviewer 1 Report

Unintentional injury is the leading cause of death and disability in children under 5. This research provides a novel approach to better understanding the factors that influence parental supervision in playgrounds by examining mobile phone device use. This research is insightful and makes an important contribution to the injury prevention literature. The authors should consider the following points to strengthen this manuscript.

  1. Consider including more detail in the methods around the types of playgrounds and environments (ie “from a large gated setting with multiple play areas to a small single ungated play area adjacent to a lake”).
  2. How many parent/child dyads were in playgrounds on average? How did the researchers ensure they accurately captured data for parents during the 20 minute block (ie was the researcher observing one parent at a time).
  3. When there was more than one child supervised, how did the authors capture data or chose which child was part of the observed dyad?
  4. Given that children have different developmental/supervision needs according to age, did supervision levels differ according to the perceived age of the child? Ie older children more likely to have intermittent supervision. Were there any observable patterns that could be reported here?
  5. Please include more detail on what was considered child injury potential and whether any injury occurred while observing.
  6. It would be helpful to know what proportion of parents had no MDU through the duration of the observation period (rather than according to each minute observed), the proportion with mixed MDU and no MDU, and only MDU during the observed period.
  7. Did the authors capture whether interviewed parents had children experience injury due to lack of supervision previously?
  8. Is there evidence to suggest the multitasking (ie using mobiles while supervising), leads to decreased supervision-or is it something that can coexist? Further while the authors cite evidence that “prolonged telephone calls or scrolling and typing on the screen resulted in decreased or no engagement with the child”, does this decrease actually result in increased injury?
  9. While I recognise the importance of constant supervision of children in reducing unintentional injury, have the authors considered the benefits of MDU? This was alluded to in the example of being able to take children to the park while maintain work, however benefits may also include allowing children free play time in a controlled environment-gated playground, while the parent has a break. The mitigation of potential risks and the interplay of these factors might be an area for consideration and future work.

Author Response

Please see attached report. Thank-you for your review.

Reviewer 2 Report

Hi. 

Thank you for all of your hard work on this paper. I think it is a solid contribution to the literature. Here are some comments that I believe will strengthen the paper.

1) The Introduction is very strong. It very clearly presents the existing literature. One thing that is needed, is a more direct link to the gaps in the existing literature. This is missing right now and would strengthen your paper.

2) Methods: "Injury Potential: ‘Decreased injury potential’ and ‘increased injury potential'" It is very difficult to understand the quality of your quantitative / observational analysis due to the fact you do not clearly define decreased and increased injury potential. Please fully define these terms and provide some examples of what you saw.

3) Methods: I believe you need to do an analysis by approximate ages. Not supervising a 2 year old is different than not supervising a 5 year old. Similarly, it would be nearly impossible to not supervise children under 1. Does that bias your data? I also would be curious on how education level / neighbourhood SES influences behaviour? I think you can do more to tease out some of these patterns.

4) Methods: Move the data collection for focus groups up into the data collection section. It should not be presented with the data analysis. Also, can you provide information about sample size, the recruitment rates, and the qualitative rigor of your data collection? Did you get saturation in answers or is your sample "under powered" for this paper? 

5) Results: The qualitative assessment needs a little more narrative around the findings. One sentence describing your finding with example quotes is no the best way to present qualitative findings. Summarize your findings and support them with quotes. Right now it is very disjoined and hard to follow. 

6) Results:  I am also curious if you found any differences among different subgroups of parents. Were there differences based on ages of children? Were there differences among males vs. females? I feel there could be some deeper evaluation of the results here.

6) Discussion: I find that you are missing a whole set of literature around risky play. Obviously that should involve supervision, but there is a whole literature suggesting that just because there is a chance for injury, doesn't mean kids shouldn't do it. I feel that providing this lens to your findings would be very helpful for readers!

7) Limitations: I believe you should mention sample size as a potential limitation. It is really difficult to tell from what is reported whether you have sufficient sample size. 

Author Response

Please find my our comments attached, thank-you for your review.

Round 2

Reviewer 2 Report

Hi,

Thank you for your responses. I appreciate your well thought out discussion. I still believe you need to make a couple changes before this is ready for publication.

1) I do not understand why Lines 172 - 175 are in Analysis. These are clearly data collection and should be moved.

2) My comment regarding "power" was really about your quantitative data collection. How does your sample compare to similar observational studies? I am not sure if you have sufficient data to reach the conclusions you are making. Please include something in your methods about this.

3) Similarly, with your qualitative paper, did you reach saturation in your data collection? This important to note to ensure qualitative rigor. 

4) Finally, my comment around gender were regarding parents. Did Dads have any different responses than Mothers? I know you oversampled mothers, but are there any comparative analysis you can do? 

Author Response

Please see attachment to reply to Reviewer 2.

Many thanks.
